# Aluminum-Doped Zinc Oxide as Front Electrode for Rear Emitter Silicon Heterojunction Solar Cells with High Efficiency

**Daniel Meza [1],\*, Alexandros Cruz [2], Anna Belen Morales-Vilches [2] , Lars Korte [1] and Bernd Stannowski [2]**

[1]     Helmholtz-Zentrum Berlin für Materialien und Energie, Institute for Silicon Photovoltaics, Kekuléstraße 5, 12489 Berlin, Germany; lars.korte@helmholtz-berlin.de

[2]     PVcomB, Helmholtz-Zentrum Berlin für Materialien und Energie, Schwarzschildstr. 3, 12489 Berlin, Germany; alexandros.cruz@helmholtz-berlin.de (A.C.); ana.morales_vilches@helmholtz-berlin.de (A.B.M.-V.); bernd.stannowski@helmholtz-berlin.de (B.S.)

\*     Correspondence: daniel.meza@helmholtz-berlin.de; Tel.: +49-30-8062-41335

**Abstract:** Transparent conductive oxide (TCO) layers of aluminum-doped zinc oxide (ZnO:Al) were investigated as a potential replacement of indium tin oxide (ITO) for the front contact in silicon heterojunction (SHJ) solar cells in the rear emitter configuration. It was found that ZnO:Al can be tuned to yield cell performance almost at the same level as ITO with a power conversion efficiency of 22.6% and 22.8%, respectively. The main reason for the slight underperformance of ZnO:Al compared to ITO was found to be a higher contact resistivity between this material and the silver grid on the front side. An entirely indium-free SHJ solar cell, replacing the ITO on the rear side by ZnO:Al as well, reached a power conversion efficiency of 22.5%.

**Keywords:** photovoltaics; silicon heterojunction; transparent conductive oxide; rear emitter

## 1. Introduction

Silicon heterojunction (SHJ) solar cells reach the highest power conversion efficiencies (η) obtained for crystalline silicon (c-Si) solar cells [1]. High open circuit voltages are obtained due to the effective passivation of defects at the surface of the c-Si absorber by thin layers of intrinsic hydrogenated amorphous silicon ((i) a-Si:H) [2,3]. Thin layers of amorphous or nanocrystalline p- and n-doped silicon deposited on top of the (i) a-Si:H layers then form selective contacts for holes and electrons, respectively. Since the lateral conductivity of these layers is too low for efficient lateral collection of the charge carriers to the metallic grid, an additional transparent conductive oxide (TCO) layer, such as indium tin oxide (ITO), is deposited on top. In addition to the charge collection, an important function of the front-side TCO layers is to act as an antireflective coating. On silicon adapted to the AM1.5 solar spectrum, this requires a thickness of 70–80 nm [4]. A general drawback of using a front TCO is its parasitic absorption at short wavelengths due to band-to-band absorption close to the bandgap energy and in the NIR due to free carrier absorption [1].

In rear emitter SHJ solar cells, the majority carrier transport to the metallic grid is carried both by the front TCO and the Si wafer, allowing the use of TCOs with lower conductivity than those needed for a front emitter configuration [3,4]. In other words, in a rear emitter SHJ solar cell, the sheet resistance ($R_{sh}$) of the front TCO plays a less critical role on the cell performance, and similar or better results than those of a cell with a front emitter could be achieved. According to simulations done by Bivour et al. [5,6], e.g., a TCO with a sheet resistance as high as 200 Ω/sq would still result in solar cells with fill factor (FF) > 80% and η > 23% with a benefit of up to + 0.5% in absolute efficiency using

the rear instead of the front emitter configuration. This broadens the range of suitable materials for the front TCO instead of the usually used ITO [4,7]. Finding a replacement for ITO could benefit the production of SHJ solar cells since indium is a scarce material. For instance, such sheet resistances are achievable, e.g., with 75-nm-thick sputtered aluminum-doped zinc oxide (ZnO:Al).

In this work, we present results of rear emitter SHJ solar cells with 75-nm-thick ZnO:Al front electrodes. Transmission line method (TLM) structures and complete solar cells were prepared and compared to otherwise identical cells from our baseline with an optimized ITO reference resulting in η > 22% [8]. Excellent optical properties of the ZnO:Al front TCO are demonstrated, and an analysis of the electrical losses due to series resistance ($R_s$) of the TCO front contact is made. The results confirm the idea that using the rear emitter configuration, a front TCO with a higher $R_{sh}$ than ITO is suitable for preparing highly efficient solar cells, and that ZnO:Al, with demonstrated η > 22%, is such a TCO.

## 2. Materials and Methods

For optical and electrical characterization, aluminum-doped zinc oxide (ZnO:Al) and tin-doped indium oxide (ITO) layers were deposited on low iron soda-lime glass (Saint-Gobain planiclear, $30 \times 30$ cm) using a pulsed DC magnetron sputtering system from Leybold Optics (A600V7). For ZnO:Al deposition, a rotatable target (99:1 wt % $ZnO:Al_2O_3$) was used and for the ITO layers, a planar target (97:3 wt % $In_2O_3:SnO_2$). The rotatable target was 600 mm long and had a diameter of between 130 mm (used) and 166 mm (new). The planar target was 600 mm long and 125 mm wide. Both materials were deposited at a nominal substrate temperature of 200°C with a substrate pre-heating time of 10 minutes before the deposition. During sputtering, the admixture of oxygen to the argon sputtering atmosphere was used to tune the stoichiometry and conductivity of the layers. The $O_2$ flow ratio $r(O_2)=(O_2)/(Ar+O_2)$ was varied between 0% and 0.74% for the ZnO:Al and 0% and 5% for the ITO deposition. The exact values of the gas flows were 269 sccm argon for the 0% oxygen sample, 268.2 sccm argon and 0.8 sccm oxygen for the 0.3% oxygen sample, 267.7 sccm argon plus 1.3 sccm oxygen for the 0.48% oxygen sample, and 267 sccm argon plus 2 sccm oxygen for the 0.74% oxygen sample. The ZnO:Al thin films were deposited using a magnetron sputtering power of 2kW, with a frequency of 40 kHz and a reverse time between pulses of 1.2 μs. The substrates were transported in front of the ZnO:Al target with a speed of 0.45 m/min.

Charge carrier concentrations and Hall mobility of the layers on glass were determined by Hall measurements in the van der Pauw geometry. Sheet resistance was measured using a four-point probe. Layer thicknesses were obtained fitting a Drude–Lorentz oscillator model to optical spectrometry measurements. All layers deposited on glass had a thickness of $98 \pm 6$ nm which, when deposited on textured substrates, results in an effective thickness of 70–80 nm. The layer thickness was measured with a DEKTAK profilometer.

Figure 1 shows the structure of the SHJ solar cells. An initially 145-μm thick n-type Cz-Si wafer was used as the base. After saw-damage etching, texturing, and RCA cleaning, the amorphous intrinsic and doped silicon layers were deposited in an AKT1600 plasma-enhanced chemical vapor deposition (PECVD) cluster tool from Applied Materials. As the front surface field and contact material to the front TCO, a nanocrystalline silicon n-type layer was used [9]. For the back emitter, an amorphous silicon p-type layer was used. TCO sputtering was done through shadow masks to define 4-cm$^2$ size cells. For all cells, the front contacts were carried out by screen printing of silver contact grids, which covered 2.7% of the cell area. For bifacial cells, the back contacts also consisted of screen-printed silver grids, whereas full area sputtered silver layers were used instead for monofacial cells.

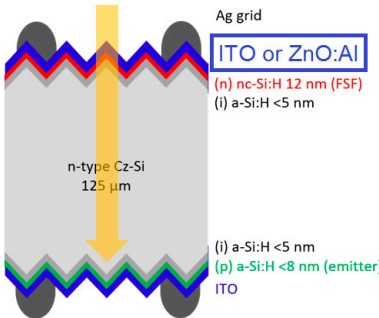

**Figure 1.** Structure of the bifacial rear emitter silicon heterojunction solar cell.

Five groups of samples (with two wafers each) were prepared. Each wafer held 14 bifacial SHJ solar cells, three transmission line method (TLM) structures, and three grid-less solar cells for EQE measurements. Four groups featured a ZnO:Al front TCO sputtered with different $O_2$ concentrations (oxygen content variation), and were named according to the oxygen partial pressure during deposition: ZnO:Al 0%, ZnO:Al 0.3%, ZnO:Al 0.48%, and ZnO:Al 0.74%. One group, with ITO as front TCO, was used as a reference and was named ITO 2.8% according to the oxygen partial pressure under which it was deposited. All of these samples had an 80-nm-thick ITO layer in the back and were processed in bifacial configuration. To test the performance of ITO-free cells, one extra group of two wafers with 14 monofacial solar cells with ZnO:Al as the front and a ZnO:Al/Ag rear contact was prepared and named MF ZnO:Al 0.3%.

The solar cells were characterized using illuminated IV measurements under AM1.5G illumination and standard test conditions in a dual source class AAA+ sun simulator. Intensity-dependent open circuit voltage (SunsV$_{oc}$) was measured using a Sinton Instruments WCT120. UV-VIS spectrometry was measured on a Perkin-Elmer LAMBDA 1050 UV/Vis/NIR Spectrophotometer. External quantum efficiency (EQE) and electrical measurements on the TLM structures were also made.

A simple calculation of the series resistance ($R_s$) of the cells was done adding the series resistance due to the transport through the TCO ($R_s^{TCO}$) and the contact resistance between the TCO and the metal grid ($R_s^c$). This is a simplified calculation of the series resistance of the cell since the contact resistivity between the TCO and the nanocrystalline silicon n-layer is neglected for these calculations, as well as the transport through the silicon wafer. Nevertheless, it represents qualitatively the electronic transport through the front stack of the solar cell in a way that allows us to understand the trends observed in the FF and j$_{sc}$ results of the previous section. $R_s^{TCO}$ and $R_s^c$ were calculated using the sheet resistance ($R_{sh}$) and contact resistivity ($\rho_c$) data extracted from the TLM measurements and the mean distance traveled by the charge carriers through the TCO and the area of the contact, respectively. The area of the contact is taken as the area of the silver grid (2.7% of the total area).

The mean distance traveled by the charge carriers through the TCO is calculated with the integral:

$$\int_0^{pitch/2} R_{sh} \cdot x\,dx$$

where "pitch" is the distance between grid fingers and $R_{sh}$ is the sheet resistance extracted from the TLM measurements. The $R_{sh}$ is extracted from the TLM measurements as the slope of the trendline added to the measured points.

$R_s^c$ is calculated using the $\rho_c$ value extracted from the TLM measurements and divided by the area of the grid, which is also the area of the contact between silver and TCO.

## 3. Results and Discussion

### 3.1. TCO Layers on Glass

Figure 2a,b show sheet resistances ($R_{sh}$) of about 100-nm-thick ZnO:Al and ITO layers on glass, as well as the absorbed light expressed as an equivalent photo-current density ($j_{loss}$). The $j_{loss}$ value is calculated by multiplying the spectrally resolved absorption of the samples with the AM1.5g spectrum, integrating over the wavelength range from 300 to 1200 nm, and subtracting the absorption of the planiclear glass. This method for estimating the absorption of the TCO layer is used due to the similarity of the samples prepared, since all of them are deposited on the same substrate with the same layer thickness and their refraction indexes and reflection are very similar. Both $R_{sh}$ and $j_{loss}$ are plotted against the oxygen partial flow during film deposition. Figure 2c shows Hall mobility ($\mu$) vs. charge carrier concentration ($N_e$) for the same layers.

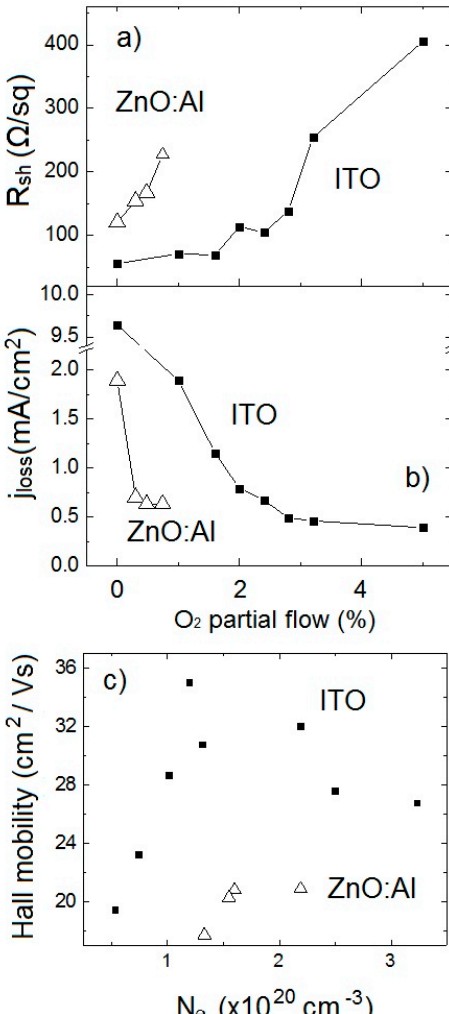

**Figure 2.** (**a**) Sheet resistance and (**b**) $j_{loss}$ due to absorption in the 300–1200 nm range of the ZnO:Al (open triangles) and indium tin oxide (ITO) (filled squares) single layers on glass deposited under different $O_2$ concentrations. (**c**) Hall mobility as a function of charge carrier concentration. Note the axis breaks in panel (**b**).

The $R_{sh}$ value of ZnO:Al layers deposited with increasing oxygen ratio in the process gas increases from 120 $\Omega$/sq to 226 $\Omega$/sq (sputtered under $r(O_2)$ = 0% and $r(O_2)$ = 0.74%, respectively).

$N_e$ decreases from $2.2 \times 10^{20}$ cm$^{-3}$ to $1.3 \times 10^{20}$ cm$^{-3}$ for the same variation of $O_2$ partial flows, which is consistent with observations by other authors [10], and is often tentatively explained by a

reduced density of oxygen vacancies in the TCO material since oxygen vacancies provide intrinsic doping on ZnO. The Hall mobility stays almost constant around 20 cm$^2$/Vs for higher $N_e$ and decreases to 18cm$^2$/Vs for the sample with the lowest $N_e$.

The $R_{sh}$, $\mu$, and $N_e$ values of the ITO layers follow a similar trend. $R_{sh}$ increases from 55 $\Omega$/sq to 405 $\Omega$/sq for an $O_2$ concentration increasing from $r(O_2)$ = 0% to $r(O_2)$ = 5%. Hall mobility maximum values of 30–35 cm$^2$/Vs are reached for layers with carrier concentrations of 1–2 $\times$ 10$^{20}$ cm$^{-3}$, decreasing for both increasing and decreasing carrier concentrations.

The Hall mobilities and charge carrier densities measured on our layers are in the same range as those reported in the literature for comparable sputtered ZnO:Al and ITO layers [10,11]. The decreasing $\mu$ with decreasing $N_e$ (for $N_e \leq 1 \times 10^{20}$ cm$^{-3}$) can be explained using the grain boundary scattering model [12]. In this model, it is assumed that the boundaries between adjacent grains or crystallites have a relatively high amount of trap states that are able to reduce the effective amount of charge carriers by trapping them. The trap states become electrically charged by this process and represent an electrical potential barrier to the free charge carriers, reducing their Hall mobility.

The increasing light absorption correlates with the increase of $N_e$ extracted from the Hall mobility measurements [13].

From these measurements and calculations, we found that ZnO:Al and ITO layers deposited under $r(O_2)$ = 0.3% and $r(O_2)$ = 2.8%, respectively, exhibit similar $R_{sh}$ (approx. 150 $\Omega$/sq) and optical absorption values (photocurrent-equivalent absorption loss 0.7 mA/cm$^2$ for the ZnO:Al and 0.5 mA/cm$^2$ for the ITO).

Figure 3 shows the $j_{loss}$ vs. sheet resistance due to the absorption of the $\approx$ 100-nm-thick ZnO:Al and ITO layers deposited on glass under the different oxygen concentrations described previously. As expected, both materials present a decrease in the $j_{loss}$ value when the sheet resistance is increased. It can be observed that the prepared ITO layers feature better optoelectronic properties than the ZnO:Al samples: for two layers with a similar $R_{sh}$, the ZnO:Al layer absorbs a larger amount of light than the ITO layers. In particular, in the region near the desired optoelectronic working point of these materials (100–150 $\Omega$/sq), the ZnO:Al layer absorbs an amount of photons equivalent to a current density between 0.6 and 2 mA/cm$^2$ while the ITO absorption stays relatively stable at a value between 0.5 and 0.8 mA/cm$^2$.

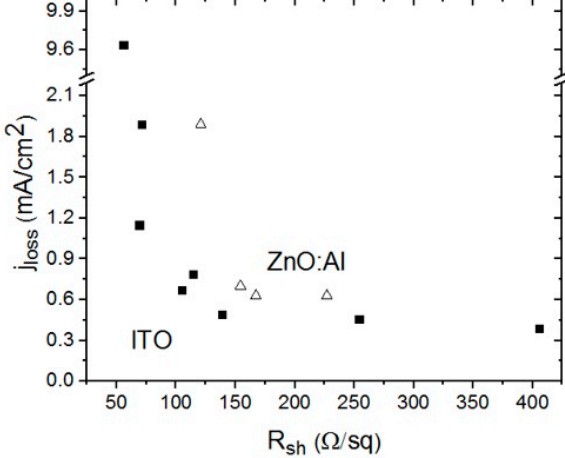

**Figure 3.** Sheet resistance vs. $j_{loss}$ due to absorption of the ZnO:Al (open triangles) and ITO (filled squares) single layers on glass. The sheet resistance values of this figure were measured with a four-point probe. The optical absorption is similar for samples with a sheet resistance between 100 and 200 $\Omega$/sq. Note the axis break in the vertical axis.

### 3.2. Transmission Line Method (TLM) Measurements

The electrical properties of TCO layers implemented in solar cells were investigated using dedicated TLM structures co-processed together with the SHJ solar cells on the same wafer. An explanation of the TLM method can be found in [14]. Figure 4 shows $R_{sh}$ and the contact resistance $\rho_c$ between the TCO layer and the silver grid. This data was extracted from TLM measurements for the front TCOs used in the SHJ structures with ZnO:Al and ITO, varying the $O_2$ concentration during the sputter deposition.

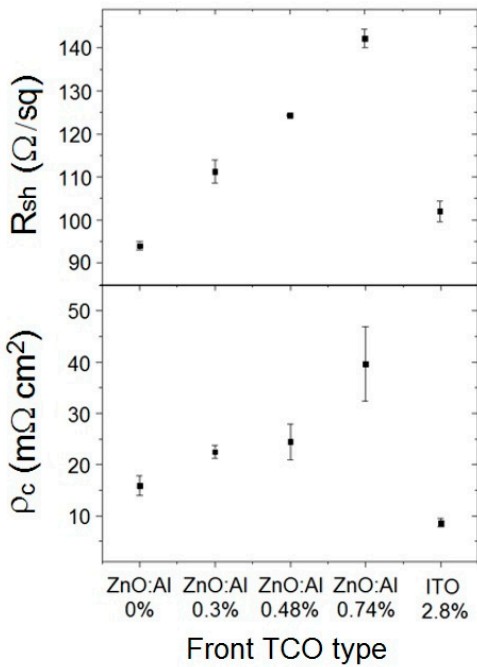

**Figure 4.** $\rho_c$ and $R_{sh}$ obtained from transmission line method (TLM) measurements on the front side of rear emitter SHJ structures with ITO 2.8% as a reference and ZnO:Al as the front TCO deposited under different oxygen partial flows. Each item on the horizontal axis represents a group of samples, with the material for the front TCO and the oxygen partial pressure under which it was deposited.

The TLM measurements show an increase in $R_{sh}$ with $O_2$ partial flow for the ZnO:Al front TCOs, which correlates with the increasing $R_{sh}$ values measured on the layers on the glass with the four-point probe (Figure 2a). It can be observed from Figure 4 that $R_{sh}$ and $\rho_c$ of the ZnO:Al TLM structures are higher than those of the ITO reference. While $R_{sh}$ of the ZnO:Al is only 5–20% higher than $R_{sh}$ of the ITO, the contact resistance $\rho_c$ is approximately three times higher. Furthermore, comparing the TCO layers on glass vs. those on the solar cells, the $R_{sh}$ value of the TLM structures increases from 94 $\Omega$/sq to 142 $\Omega$/sq for ZnO:Al with an $O_2$ partial flow ratio variation from $r(O_2) = 0\%$ to $r(O_2) = 0.74\%$, and for the ZnO:Al layers on glass it increases from 120 $\Omega$/sq to 226 $\Omega$/sq. This difference can be explained by the lateral current transport through the Si wafer.

The contact resistance $\rho_c$ of all the ZnO:Al samples with the silver grid is higher than that of the ITO reference sample, increasing from 16 m$\Omega$ cm$^2$ to 40 m$\Omega$ cm$^2$ for a partial $O_2$ flow of $r(O_2) = 0\%$ to $r(O_2) = 0.74\%$. For the ZnO:Al samples deposited under $r(O_2) = 0.3\%$ to $r(O_2) = 0.48\%$, the value is similar (around 23 m$\Omega$ cm$^2$), while the sample deposited under $r(O_2) = 0.74\%$ presents a value almost twice as large and approximately four times larger than the $\rho_c$ of the ITO reference.

Taking into account the measured $R_{sh}$ values of the ZnO:Al layers on the glass, and the wafer having a resistivity of $\approx 5$ $\Omega$ cm at 125 $\mu$m thickness ($R_{sh,cSi} = 400$ $\Omega$/sq), the equivalent sheet resistance of the system can be calculated assuming a simple model of two resistors (the TCO and the cSi) connected in parallel: $R_{sh,tot} = [(R_{sh,cSi})^{-1} + (R_{sh,TCO})^{-1}]^{-1}$. According to this model, the overall

resistance of the system then ranges from 92 $\Omega$ for a sample deposited under $r(O_2)$ = 0% to 144 $\Omega$ for a sample deposited under $r(O_2)$ = 0.74%. Note that this model neglects TCO/Si contact resistivity and changes in charge carrier density in the c-Si, which occur both upon contact formation and under illumination, i.e., excess charge carrier generation. Interestingly, the calculated values are in good accordance with the $R_{sh}$ values determined from the TLM measurements. This indicates that the assumption of a negligible TCO/Si contact resistivity is probably justified.

Using the data obtained from the TLM measurements, an estimation of the different contributions of the $R_s$ of the solar cells was made (Table 1). A direct comparison to the series resistance of the solar cells obtained when comparing the IV curve with the SunsV$_{oc}$ curve is also shown in Table 1. This series resistance is calculated following the method shown in [15].

**Table 1.** Calculated contact resistance ($R_s{}^c$), transparent conductive oxide (TCO) resistance ($R_s{}^{TCO}$) and its addition. Series resistance of the cell extracted from SunsV$_{oc}$ and IV measurements ($R_s{}^{SunsVoc\text{-}IV}$).

| TCO Variation | $R_s$ | $R_s{}^{TCO}$ | $R_s{}^c + R_s{}^{TCO}$ | $R_s{}^{SunsVoc\text{-}IV}$ |
|---|---|---|---|---|
| | [$\Omega$] | [$\Omega$] | [$\Omega$] | [$\Omega$] |
| 0% | 0.5 | 0.3 | 0.8 | 1.1 |
| 0.3% | 0.8 | 0.4 | 1.2 | 1.1 |
| 0.48% | 0.8 | 0.4 | 1.2 | 1.1 |
| 0.74% | 1.3 | 0.5 | 1.8 | 1.2 |
| ITO | 0.3 | 0.4 | 0.7 | 1.0 |

The simple model used for estimating the $R_s$ contributions of the wafer/TCO/silver grid system uses the data extracted from the TLM measurements and takes into account two main contributions: one due to lateral transport through the TCO to the silver grid and the other due to the contact between TCO and the silver grid (calculated using the $\rho_c$ extracted from the TLM measurements and dividing it through the contact area between the TCO and the silver grid).

Table 1 shows the calculated contact resistance ($R_s{}^c$), TCO resistance ($R_s{}^{TCO}$), and the sum of those two quantities. The last column shows the $R_s$ measured from the comparison of the SunsV$_{oc}$ and the sun simulator IV curve ($R_s{}^{SunsVoc\text{-}IV}$).

The estimated $R_s$ using a simple model is of the same order of magnitude as that found when comparing the IV and the SunsV$_{oc}$ measured curves. However, the changes in $R_s{}^{SunsVoc\text{-}IV}$ are less pronounced than $R_s{}^c + R_s{}^{TCO}$ expected from the model. It can be observed that the measured values and the calculated values are very similar for those samples deposited under $r(O_2)$ = 0.3% and $r(O_2)$ = 0.48%, and deviates more significantly for the samples deposited under $r(O_2)$ = 0% and $r(O_2)$ = 0.74%, which are the extremes of the oxygen variation. The layer deposited under $r(O_2)$ = 0% possess both the lowest $R_{sh}$ and the highest absorption (and therefore, smaller $j_{sc}$), while for the layer deposited under $r(O_2)$ = 0.74%, the opposite is true. It is possible that the combination of these factors (low Rsh and low $j_{sc}$ for one of the samples, high $R_{sh}$ and high $j_{sc}$ for the other) accounts for the deviation between the measured and the calculated values for these samples. This is also confirmed by the TLM measurements where it can be seen that the contact resistivity of the layer deposited under $r(O_2)$ = 0.74% to the silver grid is almost twice the value of those deposited under $r(O_2)$ = 0.3% and $r(O_2)$ = 0.48%.

Furthermore, it is important to note that the $R_s$ contribution from the lateral transport in the TCO behaves differently than $R_s$ contribution from the TCO/Ag contact resistance: The $R_s$ due to the transport stays almost constant for all the ZnO:Al samples and is similar to that of the ITO reference, while the $R_s$ due to the contact is higher for all the ZnO:Al samples than that of the ITO reference. This result was expected already from the TLM measurement analysis.

For a more accurate analysis of the $R_s$ difference between the measured values and the calculated ones, it would be useful to repeat the TLM measurements under illumination since the conductivity of the system could be different when photocurrent generation is taking place. Under illumination,

the photogenerated charge carriers would lower the effective resistivity of the TCO/c-Si stack [13], producing a smaller deviation between the calculated and measured values.

### 3.3. Solar Cell Results

Figure 5 shows the IV parameters for the bifacial SHJ solar cells with ZnO:Al and ITO as front TCOs. The TCO on the rear side of all the cells was kept constant and the reference ITO ($r(O_2) = 2.8\%$) was used. With the exception of the ZnO:Al sample deposited under 0% oxygen partial pressure, the solar cells with ZnO:Al front TCO show a slightly higher $j_{sc}$ compared to those of the ITO reference cells. The $j_{sc}$ of these solar cells increases with increasing $O_2$ partial flow. This will be further explained in the next section where the characteristics of the single layers on glass are discussed. The fill factors of all ZnO:Al samples are lower than that of the ITO sample, and they show a decreasing trend with increasing $O_2$ partial flow ($r(O_2) = 0 \ldots 0.74\%$). This result agrees qualitatively with the increase of the $R_{sh}$ value of ZnO:Al layers deposited under an increasing oxygen ratio in the process gas. The average FF of the ITO bifacial samples (78.4%) is higher than those of the ZnO:Al solar cells. This is the main reason for the better performance of the ITO solar cells, since their $j_{sc}$ and $V_{oc}$ are in the same range as those of the ZnO:Al solar cells, excluding the ZnO:Al solar cell deposited under 0% oxygen flow, which presents both a lower $j_{sc}$ and $V_{oc}$ than all other samples. The $V_{oc}$ of all these cells is very similar and in the range of approx. 728 mV.

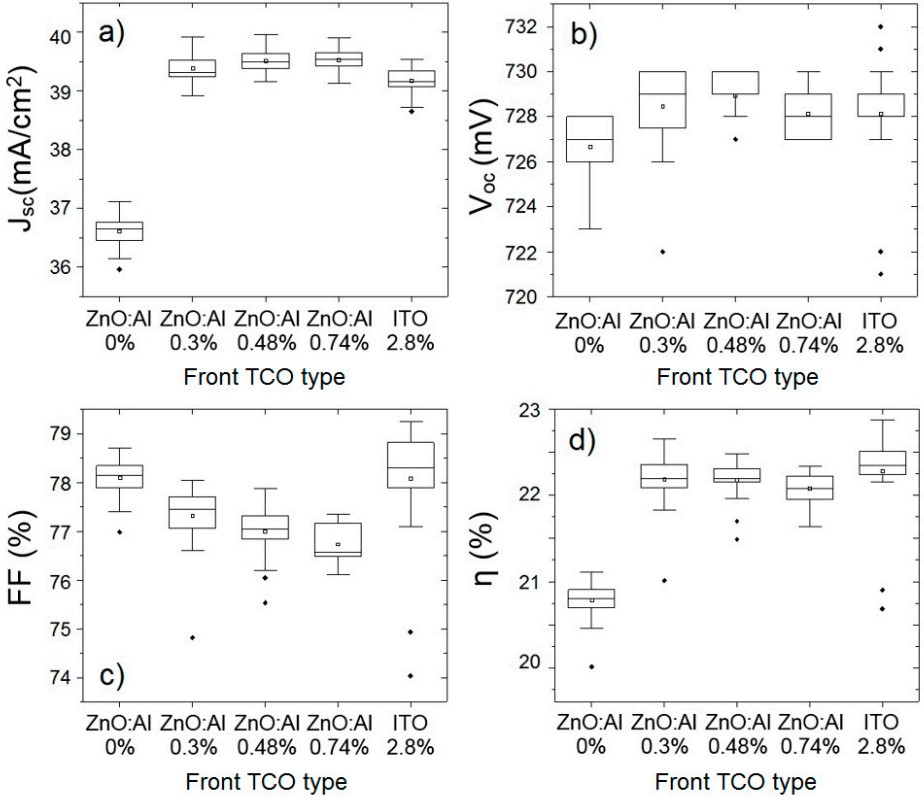

**Figure 5.** (**a**) $j_{sc}$, (**b**) $V_{oc}$, (**c**) FF and (**d**) η for the bifacial ZnO:Al solar cells and the ITO reference. The box plots represent the statistics of the measurements made on the two wafers with 14 cells each. The horizontal axis of each graph shows the material used in that group as a front TCO and the partial oxygen flow under which this material was deposited.

The smaller FF of the ZnO:Al solar cells is explained in Section 3.3 in terms of a higher value of the series resistance of the solar cells, measured using the TLM structures.

Figure 6 shows the external quantum efficiency (EQE) measurements of the solar cells with the highest power conversion efficiency of three groups: a bifacial ITO cell, and a bifacial and a monofacial

ZnO:Al solar cell both deposited under r(O$_2$) = 0.3%, as well as 1-R$_{tot}$, the fraction of non-reflected light from the UV-VIS measurements, normalized to 1.

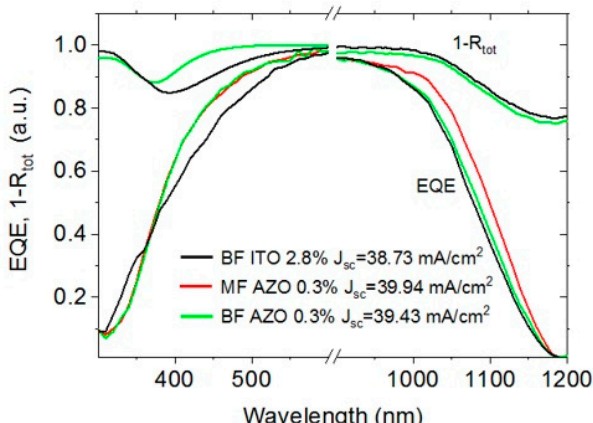

**Figure 6.** External quantum efficiency (EQE) and 1-R$_{tot}$ vs. wavelength of the solar cells with the highest power conversion efficiency of three groups. Monofacial and bifacial ZnO:Al solar cells (red and green lines, respectively) compared to a bifacial ITO reference solar cell (black line). For the 1-R$_{tot}$ spectra, only one of the ZnO:Al curves is shown since there is no difference between the monofacial and the bifacial cell on the UV-VIS reflection measurement of the front side.

The monofacial (MF) ZnO:Al solar cell has a higher EQE for long wavelengths, mainly due to the silver back contact, which reflects a fraction of the light back into the solar cell. In the case of the bifacial solar cells, a fraction of light is transmitted, lowering the j$_{sc}$.

The EQE response of both bifacial cells could suggest that the ITO cell performs worse than the ZnO:Al cell in the short wavelength region. Nevertheless, when analyzed in detail, it can be seen that the reflection minimum is different for ITO and ZnO:Al, which could be caused by a thickness deviation of the ITO layer. The j$_{sc}$ of the EQE measurements agree qualitatively with the IV measurements.

## 4. Conclusions

In this work, we compared the use of ZnO:Al films instead of the usual ITO as the front TCO in rear emitter SHJ solar cells. The results of ZnO:Al and ITO layers grown on glass with a variation in the oxygen content during the sputtering deposition demonstrate that for optimized layers with the same sheet resistance, the light absorption is very similar. As a result, the conversion efficiency of the best ZnO:Al-based solar cells of 22.6% is only slightly below that of the best ITO-based solar cells of 22.8%. Optically, the ZnO:Al front TCO even outperforms the ITO, as demonstrated by $\approx$ 0.3 mA/cm$^2$ higher j$_{sc}$s. However, it was found that the higher series resistance originating from a higher contact resistance between ZnO:Al and the screen-printed silver grid leads to a slightly lower efficiency of the ZnO:Al-based solar cells due to an increased R$_s$ and thereby lowered fill factor. These results demonstrate that ZnO:Al layers can be used in rear emitter SHJ solar cells with only a very small efficiency loss, confirming published simulation studies [5,6]. An indium-free monofacial cell achieved η = 22.5% showing that the replacement of ITO/Ag by ZnO:Al/Ag as a back contact produces cells with practically the same efficiency.

**Author Contributions:** D.M. is the main author, and he carried out the experiments, analyzed the data, planned and wrote the original draft. A.C. helped with the preparation of the TCOs, their characterization and the analysis of the results. A.B.M.-V. processed the solar cells and contributed to their characterization. B.S. conceptualized the study. L.K. and B.S. supervised the study, advised on the use of methods and data analysis, acquired funding and were responsible for project administration. All authors participated in the discussion of the results, reviewed the manuscript, and provided input for editing.

**Funding:** This research was funded by the German Ministry of Economic Affairs and Energy (BMWi) in the framework of the HERA project (reference #0325825l).

**Acknowledgments:** The authors would like to thank their colleagues of HZB for their help with the realization of this work: Katja Mayer-Stillrich and Manuel Hartig for the sputtering depositions and their useful advice. Sebastian Neubert for help with the analysis of the layers on glass and experiment planning. Max-Sebastian Hendrichs and Holger Rhein for screen printing. Luana Mazzarella and Sophie Kolb for their help with the EQE measurements. Darja Erfurt and Stefan Körner for the discussions about the results. Dorothee Menzel and Matthias Zellmeier for proofreading the manuscript.

**Conflicts of Interest:** The authors declare no conflict of interest.

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
