# Peer review of "Aluminum-Doped Zinc Oxide as Front Electrode for Rear Emitter Silicon Heterojunction Solar Cells with High Efficiency"

_applsci, doi:10.3390/app9050862_

Reviewer 1 Report

This paper describes preliminary results about the applicability of Al-ZnO thin film as a front transparent conducting electrodes for silicon heterojunction solar cell. The cell performance of Al-ZnO using solar cell was similar to that of ITO using solar cell. So, they propose that Al-ZnO can be a promising candidate for indium free TCO for solar cell. This paper can have its value from the practical engineering view point in that AZO layer was adopted for the replacement of ITO, even though the performance are slightly lower than the cell using ITO.

This reviewer has some concerns as followings,

The oxygen flow rate affected the sheet resistance of Al-ZnO thin films. They assumed that increase oxygen flow rate has suppressed the formation of intrinsic oxygen vacancies (Vo) to result in the number of free electrons generated from the vacancies. To support this assumption, there should be the Zn/O stoichiometry ratio with oxygen flow rate. So, please address what wat the exact oxygen/Zn ratio variation with oxygen flow rate.  

We have to consider are the resistivity of the thin film itself and the contact resistivity between the Al/TCO/cSi, where we have two interfaces. But, the authors have ignored the resistance in interface between TCO/cSi. Is this a reasonable assumption?  

There are ambiguities in describing the resistivity, this reviewer has experienced some difficulties to ascertain among the Rsh, Rs, Rc, R. For example, in the line 180 through 189, the expression on the resistivity or resistance of TCO or TCO/aSi, they are using the unit of ohm. Is this a correct description? So, please clarify all the element of the resistance in the cell through Ag/TCO/aSi.

 So, this reviewer recommends for the authors to revise this manuscript under consideration the above comments.

Author Response

Please see the attached document for the images.

Reply to the reviewer:

1. The oxygen flow rate affected the sheet resistance of Al-ZnO thin films. They assumed that increase oxygen flow rate has suppressed the formation of intrinsic oxygen vacancies (Vo) to result in the number of free electrons generated from the vacancies. To support this assumption, there should be the Zn/O stoichiometry ratio with oxygen flow rate. So, please address what wat the exact oxygen/Zn ratio variation with oxygen flow rate.

Reply: For this study we considered the oxygen partial pressure as the varying parameter that changes the properties of our thin films. For this reason we didn´t realize any XPS measurements that allow us to determine the stoichiometry of the samples. This analysis would be of course useful for understanding the amount of oxygen vacancies in the material, nevertheless, the increase of the charge carrier concentration with a decrease of the oxygen partial flow is also an evidence of the way in which the oxygen vacancies reduce the sheet resistance of this thin films.

The following figure shows this trend, in case the reviewer would like to review it, even though the data are also presented in figure 2.

2. We have to consider are the resistivity of the thin film itself and the contact resistivity between the Al/TCO/cSi, where we have two interfaces. But, the authors have ignored the resistance in interface between TCO/cSi. Is this a reasonable assumption?  

Reply: The idea behind calculating a simplified system where the contact resistance of the interface TCO/c-Si is ignored is due to the analysis of the resistance values showed in table 1. It can be seen, that the difference between the calculated values and the measured ones (for all the samples except for the extreme one deposited under 0.74% oxygen partial flow) is smaller than Rsc and of the same order of magnitude than the RsTCO (between 0.1 and 0.3 Ohm). If we suppose that the difference between measured and calculated value of the resistance is due to the contact resistance between TCO and c-Si, it is a reasonable assumption to ignore it, since we propose, that the main responsible of the bad performance of the ZnO:Al is the contact resistance between TCO and silver grid.

3. There are ambiguities in describing the resistivity, this reviewer has experienced some difficulties to ascertain among the Rsh, Rs, Rc, R. For example, in the line 180 through 189, the expression on the resistivity or resistance of TCO or TCO/aSi, they are using the unit of ohm. Is this a correct description? So, please clarify all the element of the resistance in the cell through Ag/TCO/aSi.

Reply: It is explained now in the text but also here.

It can be somewhat confusing indeed, that these two quantities have the same unit, nevertheless, both are calculated in a different way, which gives as a result the unit Ohms. For the case of the contact resistance, it is calculated as a contact resistivity value (with units equal to Ohm cm^2) divided by the area of the contact (with units equal to cm^2), producing a result in Ohm units.

The series resistance due to the TCO (RsTCO) are calculated using the sheet resistance value (Rsh with units equal to Ohm/sq) multiplied by the average mean path that the charge carriers “travel” through the TCO, producing also a result in Ohm units.

Reviewer 2 Report

1. The detailed sputtering parameters for the deposition of TCO films should be provided.

2. Line 109, How were the thicknesses of the films determined?

3. Line 111, for the spectrally resolved absorption of the samples, how the absorption is determined?

4. Line 141-142, the absorption spectra of the as-deposited TCO should be given in order to better understand the description in line 141 and jloss depicted in Fig. 3.

5. Line 103 and Line 204, how the TCO resistance (RsTCO) and the contact resistance (Rsc) were calculated? The method should be described, and the references of the method should be cited.

6. For Fig. 2, the plots of carrier density versus oxygen partial flow and Hall mobility versus oxygen partial flow should be given in order to clearly understand the variation trend of carrier density and Hall mobility.

7. Line 118-119, for Fig. 2a, please explain why the sheet resistance decreased with an increase in the oxygen flow.

8. Line 163, how were Rsh and the contact resistance ρc between the TCO layer and the silver grid determined? The method should be given, and the references of the method should be cited.

9. For Fig. 4 and Fig. 5, the label for x-axis should be given.

10. Line 186-187, to determine the sheet resistance of the system, please clearly define the terminals, the resistors in parallel, and the system. And describe the measurement method clearly.

11. Line 205-206, How was Rs determined from the comparison of the SunsVoc and the sun simulator IV curve (RsSunsVoc-IV)? The method should be described, and the references of the method should be cited.

Author Response

Please see the attached file for the images

Reply to the reviewer:

1. The detailed sputtering parameters for the deposition of TCO films should be provided.

The detailed parameters of the sputtering process have been written now in the text.

2. Line 109, How were the thicknesses of the films determined?

Using a DEKTAK profilometer, the information is included now in the text.

3. Line 111, for the spectrally resolved absorption of the samples, how the absorption is determined?

First the amount of reflected and transmitted light are subtracted from 100% and that is considered the absorption spectrum. The absorption spectrum is then multiplied with the AM1.5 spectrum to calculate the equivalent photocurrent that the layers are absorbing.

A substrate without a TCO layer is also measured with the UV-VIS spectrometer and the amount of light absorbed by the substrate alone is taken as a standard and subtracted from the values of the TCO layers.

4. Line 141-142, the absorption spectra of the as-deposited TCO should be given in order to better understand the description in line 141 and jloss depicted in Fig. 3.

In the opinion of the authors, the absorption of the spectra could distract the reader from the continuity of the paper since we are only analyzing the TCOs as a part of the finished device. The absorption spectra of these samples are very similar and only small differences can be seen between them. Nevertheless, when multiplied with the AM 1.5 spectrum, a single number is produced, which we call jloss, and is easy to compare between samples.

The absorption spectra are shown in the next figure, for the reviewer to analyze if he´d like to.

5. Line 103 and Line 204, how the TCO resistance (RsTCO) and the contact resistance (Rsc) were calculated? The method should be described, and the references of the method should be cited.

The calculation of RsTCO  and Rsc, are relatively simple calculations, I included now in the text, but also a short explanation here:

It can be somewhat confusing indeed, that these two quantities have the same unit, nevertheless, both are calculated in a different way, which gives as a result the unit Ohms. For the case of the contact resistance, it is calculated as a contact resistivity value (with units equal to Ohm cm^2) divided by the area of the contact (with units equal to cm^2), producing a result in Ohm units.

The series resistance due to the TCO (RsTCO) are calculated using the sheet resistance value (Rsh with units equal to Ohm/sq) multiplied by the average mean path that the charge carriers “travel” through the TCO, producing also a result in Ohm units.

6. For Fig. 2, the plots of carrier density versus oxygen partial flow and Hall mobility versus oxygen partial flow should be given in order to clearly understand the variation trend of carrier density and Hall mobility.

This is of course a sensible idea, nevertheless, we thought that the reader could be bored by showing two graphs that contain expected trends for these parameters. For this reason we condensed the Hall mobility and carrier density information of the samples in the figure 2c), which leads to a better understanding (in our opinion) of the optoelectronical properties of these layers.

If the reviewer considers these two graphs are absolutely necessary they could be added, and i´ll add them in this review document for her/him to analyze them if she/he wants.

7. Line 118-119, for Fig. 2a, please explain why the sheet resistance decreased with an increase in the oxygen flow.

If Fig. 2a is analyzed in detail it can be seen that this is not the case. The sheet resistance increases with an increase in oxygen flow.

8. Line 163, how were Rsh and the contact resistance ρc between the TCO layer and the silver grid determined? The method should be given, and the references of the method should be cited.

Rsh was determined using a four point probe which is a common measuring technique.

The ρc and Rsh (from figure 4) between the TCO layer and the silver grid using TLM structures,as explained in the following reference:

S. Luan, G.W. Neudeck, J. Appl. Phys. 72, 766 (1992)

This reference is also added now in the text

The information about which values were measured with which method is now also added in the text.

9. For Fig. 4 and Fig. 5, the label for x-axis should be given.

The label was added to the figures.

10. Line 186-187, to determine the sheet resistance of the system, please clearly define the terminals, the resistors in parallel, and the system. And describe the measurement method clearly.

The sheet resistance is extracted from the measurements made with the TLM structures and it’s the slope of the trendline that can be added to the measurement points (i´ll add this info to the text in the materials and methods section). This is also explained in the reference:

S. Luan, G.W. Neudeck, J. Appl. Phys. 72, 766 (1992)

In my opinion the system is also described in lines 194-199. It is a very simple system, it only consists of two resistors in parallel, since we are neglecting the contact resistance between TCO and c-Si. The terminals are the silver pads of the TLM structures.

11. Line 205-206, How was Rs determined from the comparison of the SunsVoc and the sun simulator IV curve (RsSunsVoc-IV)? The method should be described, and the references of the method should be cited.

This was calculated with the method shown in section 2.3 of the following reference:

Pysch et al., Solar Energy Materials & Solar Cells 91 (2007) 1698–1706

This information and reference are now also added in the text.

Round  2

Reviewer 1 Report

Revised manuscript have conveyed a proper explanations. Thank you.